# Genome-Wide Identification and Characterization of the *TIFY* Gene Family and Their Expression Patterns in Response to MeJA and Aluminum Stress in Centipedegrass (*Eremochloa ophiuroides*)

**DOI:** 10.3390/plants13030462

**Published:** 2024-02-05

**Authors:** Haoran Wang, Yuan Zhang, Ling Zhang, Xiaohui Li, Xiang Yao, Dongli Hao, Hailin Guo, Jianxiu Liu, Jianjian Li

**Affiliations:** 1The National Forestry and Grassland Administration Engineering Research Center for Germplasm Innovation and Utilization of Warm-Season Turfgrasses, Institute of Botany, Jiangsu Province and Chinese Academy of Sciences, Nanjing Botanical Garden, Memorial Sun Yat-Sen, Nanjing 210014, China; njlydxwhr@163.com (H.W.); hndongli@163.com (D.H.);; 2Jiangsu Key Laboratory for the Research and Utilization of Plant Resources, Institute of Botany, Jiangsu Province and Chinese Academy of Sciences, Nanjing Botanical Garden, Memorial Sun Yat-Sen, Nanjing 210014, China

**Keywords:** *Eremochloa ophiuroides*, TIFY, genome-wide, aluminum stress, phylogenetic analysis, hormone, expression pattern

## Abstract

The TIFY family is a group of novel plant-specific transcription factors involved in plant development, signal transduction, and responses to stress and hormones. *TIFY* genes have been found and functionally characterized in a number of plant species. However, there is no information about this family in warm-season grass plants. The current study identified 24 *TIFY* genes in *Eremochloa ophiuroides*, a well-known perennial warm-season grass species with a high tolerance to aluminum toxicity and good adaptability to the barren acidic soils. All of the 24 *EoTIFYs* were unevenly located on six out of nine chromosomes and could be classified into two subfamilies (ZIM/ZML and JAZ), consisting of 3 and 21 genes, respectively, with the JAZ subfamily being further divided into five subgroups (JAZ I to JAZ V). The amino acids of 24 EoTIFYs showed apparent differences between the two subfamilies based on the analysis of gene structures and conserved motifs. MCScanX analysis revealed the tandem duplication and segmental duplication of several *EoTIFY* genes occurred during *E. ophiuroides* genome evolution. Syntenic analyses of *TIFY* genes between *E. ophiuroides* and other five plant species (including *A. thaliana*, *O. sativa*, *B. distachyon*, *S. biocolor*, and *S. italica*) provided valuable clues for understanding the potential evolution of the EoTIFY family. qRT-PCR analysis revealed that *EoTIFY* genes exhibited different spatial expression patterns in different tissues. In addition, the expressions of *EoTIFY* genes were highly induced by MeJA and all of the EoTIFY family members except for *EoJAZ2* displayed upregulated expression by MeJA. Ten *EoTIFY* genes (*EoZML1*, *EoZML1*, *EoJAZ1*, *EoJAZ3*, *EoJAZ5*, *EoJAZ6*, *EoJAZ8*, *EoJAZ9*, *EoJAZ10*, and *EoJAZ21*) were observed to be highly expressed under both exogenous MeJA treatment and aluminum stress, respectively. These results suggest that *EoTIFY* genes play a role in the JA-regulated pathway of plant growth and aluminum resistance as well. The results of this study laid a foundation for further understanding the function of *TIFY* genes in *E. ophiuroides*, and provided useful information for future aluminum tolerance related breeding and gene function research in warm-season grass plants.

## 1. Introduction

Centipedegrass (*Eremochloa ophiuroides* (Munro) Hack.) is a perennial warm-season (C_4_) grass species of the *Eremochloa* genus of the Poaceae family, which is indigenous to China and is now widely distributed in the Yangtze River Basin and its southern area in China, Southeast Asia, with a widespread application in East Asia and the United States [1,2]. *E. ophiuroides* has highly developed stolons and a typical prostrate growth habit in the field, and it is mainly used as turf for home lawns and recreational fields and for soil conservation in tropical and subtropical regions. As a native species in southern China, *Eremochloa ophiuroides* is naturally distributed in the acid soil areas and is well known for its strong tolerance to aluminum (Al) toxicity and good adaptation to infertile soils. In particular, with the implementation of ecological civilization construction in China, *E. ophiuroides* has been showing great potential for land surface greening in acid soil areas. In view of the high tolerance to Al toxicity and good growth in both hydroponic solution and soil culture, *E*. *ophiuroides* could be used a model of warm-season grasses for exploring mechanisms of Al tolerance [3].

TIFY, previously known as ZIM (zinc finger protein expressed in the inflorescence meristem), is a novel plant-specific gene family coding for transcription factors with a diversity of functions in plant development, signal transduction, and responses to stress and hormones. This family was firstly identified from *Arabidopsis thaliana* (*A. thaliana*) and was found to have a highly conserved TIFY motif (TIF[F/Y]XG) in the protein sequences [4]. The TIFY family is generally comprised of four major groups in dicots or three in monocots, including the Zinc-finger expressed in Inflorescence Meristem (ZIM)/ZIM-like (ZML), TIFY (absent in monocots), PEAPOD (PPD), and JAZ groups [5]. Except for the TIFY group containing only the TIFY domain, the other three groups have at least one other structural domain. ZIM/ZML group contains a C2C2-GATA zinc-finger DNA-binding domain and a CCT domain (CONSTANS, CO-like, TOC1), the PPD group has a Jas domain defecting PY-NLS motif (jasmonate-associated) and a PPD domain, while the JAZ group contains a C-terminal jasmonate acid (JA)-associated (Jas, CCT-2) domain, which interacts with the MYC2 proteins to inhibit the JA signaling pathway [5,6,7].

A growing number of studies have shown that TIFY family genes participate in many diverse biological processes in plants. *TIFY* genes play a remarkable role in plant growth and developmental processes. For instance, *ZIM*/*TIFY1* [8] and *TIFY4b* (*PPD2*) [9] were found to be involved in petiole and hypocotyl elongation and leaf development in *Arabidopsis*, respectively. *OsTIFY11b* (*OsJAZ10*) and *OsTIFY3* (*OsJAZ1*) are regulators involved in grain size and spikelet development in rice [10,11]. In addition, TIFYs are broadly involved in plant responses to biotic stresses such as *Pseudomonas syringae DC3000* (jaz10 mutants) [12] and bacterial blight resistance (*OsJAZ8*) [13] and multiple abiotic stresses. The overexpression of *TaJAZ1* could increase bread wheat powdery mildew resistance [14]. Overexpressed *GhJAZ2* led to increased sensitivity to salt stress in transgenic cotton [15]. Upregulation of the *AtTIFY10a*, *10b*, *GsTIFY10a*, and *OsJAZ8* genes under alkaline and salt stresses indicated that these genes might be involved in the response to abiotic stresses [16,17]. Moreover, some TIFY family members are involved in modulating the signaling pathways of hormones such as JA and abscisic acid (ABA) [18,19]. All of these results reveal that the TIFY family genes have diverse regulatory roles in plant signal transduction and the regulation of biotic and abiotic stress responses and so might be multiple-stress-responsive genes. Interestingly, a more recent report on the involvement of TIFY family genes against Al stress revealed their potential roles in response to Al toxicity [20], while the current knowledge of *TIFY* genes in Al tolerance is very limited.

Due to the important roles of the TIFYs in diverse biological processes and stress defenses, the TIFY family has gained significant attention in the field of gene function research, especially in mining stress-resistance genes in plants. The TIFY family has recently been identified on a genome-wide level and characterized in various plant species, including wheat [21,22], rice [23], maize [24,25], and sorghum [26,27]. Yet, it has not been mentioned to date in the identification and functional characterization of *TIFY* genes in *E. ophiuroides*. At present, a high-quality chromosome-scale genome assembly of *E. ophiuroides* has been released [28], which is helpful in identifying and characterizing *E. ophiuroides TIFY* genes. In this study, to investigate the potential roles of *E. ophiuroides* TIFY proteins in response to phytohormones and Al stress, a genome-wide identification of the TIFY family genes was performed from the assembled *E. ophiuroides* genome. All of the identified *E. ophiuroides TIFY* genes were subsequently subjected to structural and phylogenetic analysis, conserved domain alignment, and investigations of the chromosomal location and syntenic relationships. Furthermore, the expression patterns of the *E. ophiuroides TIFY* genes in different tissues’ responses to phytohormone MeJA treatment and Al toxicity were examined, respectively. These results provide a foundation for further functional research of TIFY family members in *E. ophiuroides*, thus facilitating the cultivation of Al-resistant cultivars in grass plants.

## 2. Results

### 2.1. Identification of TIFY Family Genes in the E. ophiuroides Genome

In total, 24 *TIFY* genes in the whole genome of *E. ophiuroides* were identified using two BLAST methods (Appendix A), and these were renamed *EoZML1* to *EoZML3* and *EoJAZ1* to *EoJAZ21* according to their chromosomal positions and the homologous relationships with the *TIFY* genes in *Arabidopsis* and rice. The basic characteristics of these genes were analyzed, including the coding sequence (CDS) length, protein sequence length, relative molecular weight, isoelectric point, and subcellular localization prediction (Appendix A). Among the 24 EoTIFYs, the shortest protein sequence was EoJAZ18 with only 68 amino acids. The longest protein sequence was EoJAZ3, comprising 424 amino acids. In general, the shorter the amino acid sequence, the smaller the relative molecular weight of the protein. The relative molecular weight of the 24 EoTIFY proteins ranged from 7.58 KDa (EoJAZ18) to 44.47 KDa (EoJAZ3). The isoelectric points of the 24 EoTIFY proteins varied widely, ranging from 4.75 (EoZML2) to 9.94 (EoJAZ13). Interestingly, the isoelectric points of most EoTIFY proteins (20/24) were greater than seven, suggesting that EoTIFY proteins were biased towards being rich in basic amino acids. Subcellular localization prediction showed that 23 EoTIFY proteins were localized in the nucleus, and the other one was in the chloroplast.

### 2.2. Multiple Sequence Alignment, Phylogenetic Analysis, and Classification of EoTIFY Proteins

To explore the evolutionary relationship of the identified *E. ophiuroides TIFY* gene family, the amino acid sequences of 24 EoTIFY proteins together with 18 *A. thaliana* and 20 *O. sativa* TIFY proteins were used to construct a phylogenetic tree (Figure 1). According to the classification method of TIFY proteins previously reported in *A. thaliana* [4] and *O. sativa* [23], 24 *E. ophiuroides* TIFY proteins were classified into two subfamilies, namely the subfamilies ZIM/ZML and JAZ. Between the two subfamilies, the subfamily JAZ contained more members with 21 EoTIFYs, and it could be further divided into five subgroups, i.e., JAZ I to JAZ V. Of the five JAZ subgroups, JAZ V was the largest subgroup with 11 EoTIFY proteins, followed by the JAZ II subgroup with 4 members, while each of the JAZ I, JAZ III, and JAZ IV subgroups contained 2 EoTIFY members (Figure 1 and Appendix A). In addition, phylogenetic relationships among *E. ophiuroides* TIFY family members were analyzed based on performing a phylogenetic tree using only the alignment of the 24 EoTIFYs characterized herein, and the same group-derived TIFY proteins tended to be clustered together (Figure 2A). This was consistent with that of the above constructed phylogenetic tree from *E. ophiuroides*, rice, and *Arabidopsis* TIFY sequences (Figure 1).

### 2.3. Gene Structures and Conserved Motifs of EoTIFY Gene Family

In order to understand the structural diversity of *E. ophiuroides TIFY* genes, the exon–intron structure of all the identified *EoTIFY* genes was investigated. As shown in Figure 2B, *E. ophiuroides TIFY* genes showed great differences in the number of exons and the length of introns among members of the different subfamilies and subgroups, with the number of exons ranging from one to eight unevenly (Figure 2B and Appendix A). The subfamily ZIM/ZML had the largest number of exons (Figure 2B and Appendix A), with an average number >7, of which EoZML1 contained 7 exons, while both EoZML2 and EoZML3 had 8 exons. The subfamily JAZ had a divergent number of exons, with most members of the subgroup JAZ V containing one to three exons but the members of the other JAZ subgroups possessing three to seven exons except for EoJAZ2.

The conserved structural motifs were further analyzed to understand the structural and functional characteristics of EoTIFY proteins. MEME motif analysis identified a total of 10 motifs from 24 EoTIFYs, named Motifs 1 to 10 (Figure 2C and Appendix A). As shown in Figure 2C, no member of the *E. ophiuroides* TIFY family was found to contain a complete set of 10 conserved motifs, and the motif number of the EoTIFY proteins ranged from 1 to 5. Of the 10 conserved motifs, only Motifs 1 and 2 widely existed in most EoTIFY members, except for EoJAZ18 missing Motif 1 and EoJAZ12 and EoJAZ21 lacking Motif 2. In addition, some motifs were distributed only in specific families or groups. The conservative Motifs 4 and 5 were unique to the subfamily ZML; Motif 10 was unique to the subgroup JAZ I. The similar motif arrangements of the EoTIFY proteins in the same subfamily or subgroup indicated that their protein structure was conserved.

To further understand the conservation patterns of TIFY and Jas domains in different subfamilies and subgroups, their sequences were aligned and the corresponding logos were generated as shown in Figure 3A,B. The TIFY domain logo revealed that the TIFY domains were not completely conserved, but most of them shared common motifs, such as TIXYXG, TXFYNG, and TLX2QG. Among these conserved motifs, the motif TIXYXG was shared by the subgroup members of JAZ I, JAZ III, and JAZ IV, and one member of JAZ II and most members of JAZ V; the motifs of TXFYNG were shared by three JAZ II members; while the motif TLX2QG was shared by the ZML members. TIXYXG was the most dominant motif, and 16 out of 24 EoTIFY family members contained this motif (Appendix A). Compared with the TIFY domain, the Jas domain was more conserved and shared more conserved residues at the motif of SLX2FX2KRX2R (Figure 3A).

### 2.4. Chromosomal Distribution and Gene Duplication Analyses of the EoTIFY Genes

According to the genome annotation of *E. ophiuroides*, of the 24 identified *E. ophiuroides TIFY* genes, 22 genes were unevenly mapped to six out of nine chromosomes of *E. ophiuroides*, and 2 genes (*EoZML1* and *EoJAZ21*) were situated on contigSS_243 and contigSS_51_F_2_D (Appendix A). Among the TIFY-gene-distributed chromosomes, Chr1 had the most genes distributed with 13 EoTIFY members, followed by Chr2 with 4 EoTIFYs and Chr8 with 2 EoTIFYs, and Chr4, Chr5, and Chr9 with only 1 EoTIFY member each (Figure 4). 

The results of an MCScanX analysis indicated that several gene duplication events occurred throughout the *E. ophiuroides* TIFY family. The gene pairs of *EoJAZ12*/*EoJAZ13*, *EoJAZ12*/*EoJAZ14*, *EoJAZ12*/*EoJAZ15*, *EoJAZ13*/*EoJAZ14*, *EoJAZ13*/*EoJAZ15*, and *EoJAZ14*/*EoJAZ15* in the EoTIFY family were detected as tandem duplication genes. In addition to the tandem duplication event, four segmental duplication events were also identified in the EoTIFY family, which included the gene pairs of *EoJAZ6*/*EoJAZ7*, *EoJAZ11*/*EoJAZ16*, *EoJAZ10*/*EoJAZ17*, and *EoJAZ7*/*EoJAZ10* (Figure 5 and Appendix A). In order to investigate the potential selective pressure of the identified duplication gene pairs, the Ka/Ks ratios were also calculated in this study. All the Ka/Ks values of the above detected tandemly and segmentally duplicated *EoTIFY* gene pairs were less than one (Appendix A), suggesting that the repetitive *TIFY* genes in *E. ophiuroides* were primarily constrained by intense purification selection pressure.

### 2.5. Synteny and Evolutionary Analyses of the E. ophiuroides TIFY Genes and Other Plants TIFYs

To explore the potential evolutionary clues of the *E. ophiuroides TIFY* gene family, a collinear map of *E. ophiuroides* and five representative species was constructed, including four monocots plants (*O. sativa*, *B. distachyon*, *S. biocolor*, and *S. italica*) and a dicots plant (*A. thaliana*). As shown in the comparative syntenic maps (Figure 6), syntenic gene pairs were detected more frequently between *E. ophiuroides* and monocots than between *E. ophiuroides* and *A. thaliana*. A total of 24 *EoTIFY* genes showed syntenic relationships with those in *S. biocolor*, followed by *O. sativa* and *S. italica* each with 16 *EoTIFY* genes, *B. distachyon* with 12, and *A. thaliana* with 3 *EoTIFY* genes. Among the orthologous gene pairs, two *EoTIFY* genes of *EoJAZ7* (evm.model.ctg185.47) in the syntenic analysis of *E. ophiuroides* and three species of *O. sativa*, *S. biocolor*, and *S. italic*, and *EoJAZ10* (evm.model.ctg286.50) in the syntenic analysis of *E. ophiuroides* and *S. biocolor* were identified to be associated with at least three syntenic gene pairs. In addition, the Ka/Ks ratios of all the above detected collinear gene pairs were less than 1, indicating that the *E. ophiuroides TIFY* gene family might have suffered strong selection pressure during their evolution (Appendix A).

### 2.6. Expression Patterns of TIFY Family Genes in Various Tissues of E. ophiuroides

The expression patterns of the EoTIFY family genes in various tissues (root, stem, leaf, inflorescence, stolon node, and callus) were investigated by qRT-PCR analysis (Figure 7, Appendix A, and Appendix A). For the JAZ II subgroup (*EoJAZ20*, *EoJAZ2*, *EoJAZ5*, and *EoJAZ21*), a higher expression level of *EoJAZ5* and *EoJAZ21* was found in leaf, inflorescence, root, and stem tissues. However, the genes *EoJAZ20* and *EoJAZ2* showed distinct tissue expression patterns, with the *EoJAZ2* gene having the highest expression in inflorescences, but *EoJAZ20* being highly expressed in callus and root tissues. Although there are only two members in the JAZ I subgroup (*EoJAZ3* and *EoJAZ4*), a different tissue expression pattern were detected for them. The *EoJAZ3* was highly expressed in stem, leaf, and inflorescence tissues, but the *EoJAZ4* only showed the highest expression in inflorescences. For the ZML subgroup, all the three genes (*EoZML1*, *EoZML2*, and *EoZML3*) showed significantly tissue-specific expression in leaves and were expressed more highly in inflorescences than in other tissues. A similar leaf-specific expression pattern could also be found in the JAZ IV (*EoJAZ1*/*EoJAZ8*) and JAZ III (*EoJAZ6*/*EoJAZ7*) subgroup genes; and meanwhile, the *EoJAZ1*, *EoJAZ8*, and *EoJAZ6* genes showed relatively low levels of expression in stems, and *EoJAZ8* also exhibited a relatively low expression in stolon nodes. For the JAZ V subgroup, the tissue expression patterns of all gene members were diverse. Five genes (*EoJAZ16*, *EoJAZ9*, *EoJAZ17*, *EoJAZ14*, and *EoJAZ13*) displayed tissue-specific expression only in inflorescences, while the three genes *EoJAZ11*, *EoJAZ15*, and *EoJAZ18* showed the highest expression in inflorescence, leaf, and root tissues, respectively. As for *EoJAZ19* and *EoJAZ10*, although both genes were highly expressed in stems, *EoJAZ19* had the highest expression level in leaves and *EoJAZ10* showed the highest level of expression in stem tissues. In addition, except for *EoJAZ18* presenting the highest level of expression in root tissues, the genes *EoJAZ11*, *EoJAZ12*, and *EoJAZ15* were also detected at relatively low levels of expression in root tissues. Specifically, the genes *EoJAZ12* and *EoJAZ18* were also detected as highly expressed in callus tissues.

### 2.7. Expression Patterns of the EoTIFY Family Genes under MeJA Treatment

Phytohormones play vital roles in enhancing the ability of plants to adapt to harsh environmental conditions [29]. *TIFY* family genes have been found to act as key regulators of jasmonate signaling in many plant species [6,23,30,31]. Therefore, the responses of *TIFY* genes to MeJA in *E. ophiuroides* were investigated in the present study (Figure 8, Appendix A, and Appendix A). After exogenous MeJA treatment in *E. ophiuroides*, the expression of all *EoTIFY* genes were upregulated at a certain stage of the treatment or during the whole process, except for *EoJAZ2* that was decreased gradually from 1.5 h to 6 h, followed by a little increase at 24 h. In response to MeJA treatment, all three members of the ZML subfamily showed a downregulated expression level at early time points (1.5 h, 3 h, and 6 h) but were significantly upregulated at 24 h compared to the control. In subfamily JAZ, 12 out of the 20 upregulated gene members, including two members in the subgroup JAZ II (*EoJAZ5* and *EoJAZ21*), both members in the subgroup JAZ III, one member in JAZ IV (*EoJAZ8*), and seven members in JAZ V (*EoJAZ9* to *EoJAZ17* except for *EoJAZ10* and *EoJAZ12*) had the highest levels of expression at 3 h MeJA treatment. As for the other eight JAZ subfamily genes, three members in different subgroups (*EoJAZ1*, *EoJAZ4*, and *EoJAZ20*) showed gradually increasing expression from 1.5 h to 6 h and reached the highest upregulation at 6 h treatment; two members including *EoJAZ3* and *EoJAZ12* had the highest expression levels at 24 h treatment, with *EoJAZ3* displaying an increasing trend over the whole process but *EoJAZ12* showing downregulation before 24 h; while the expressions of the other three JAZ V members (*EoJAZ10*, *EoJAZ18*, and *EoJAZ19*) were upregulated only at the early stage of treatment (1.5 h). Undoubtedly, all of the *EoTIFY* family genes could be strongly induced by MeJA.

### 2.8. Expression Patterns of the EoTIFY Family Genes in Response to Aluminum Toxicity

The expression pattern of the *EoTIFY* family genes was also investigated under Al toxicity at three time points, i.e., before Al treatment (0 h), after a short time (6 h), and after a long time (24 h) of treatment. As shown in Figure 9, Appendix A, and Appendix A, members of the subfamily ZML showed different expression patterns in response to Al toxicity. *EoZML1* showed a gradually increasing expression during the whole Al treatment from 0 to 24 h, while *EoZML2* had a higher expression level only at 24 h of Al treatment. On the contrary, the expression level of *EoZML3* decreased obviously under the Al stress. In the subfamily JAZ, only 8 JAZ genes out of 21 were found to be upregulated in response to Al treatment. Specifically, one JAZ gene in each of the two subgroups JAZ I (*EoJAZ3*) and JAZ III (*EoJAZ6*), and two JAZ genes in each of the three subgroups JAZ II (*EoJAZ5* and *EoJAZ21*), JAZ IV (*EoJAZ1* and *EoJAZ8*), and JAZ V (*EoJAZ9* and *EoJAZ10*) exhibited obviously upregulated expression levels at a specific time point or during the whole stage of the Al stress, while the other thirteen JAZ genes from four subgroups including JAZ I, JAZ II, JAZ III, and JAZ V were downregulated during the whole period of or at a specific point of Al treatment. These highly induced TIFY family members, especially upregulated members in response to Al toxicity, might be good candidates for developing resistance against Al stress in *E. ophiuroides*.

## 3. Discussion

The TIFY family is an important plant-specific transcription factor family identified in both monocot and eudicot plant species [4]. A growing body of research has shown that the genes of the TIFY family, especially the members of the JAZ subfamily, play important roles in multiple plant developmental processes and in response to biotic and abiotic stresses [32,33]. As one of the significant warm-season eco-grass and turfgrass plants, *E. ophiuroides* has been frequently challenged by various biotic or abiotic stresses over its entire growth period and evolved a series of resistance mechanisms. However, the information on the TIFY family in *E. ophiuroides* is completely lacking, which is hampering the progress in exploring its resistance mechanisms. Rapid advances in sequencing technology and bioinformatic tools have dramatically increased the unprecedented ability to perform a series of genomic studies at the genome-wide scale. The release of a chromosome-scale genomic sequence of *E. ophiuroides* enables us to conduct a comprehensive analysis of the TIFY family.

In this study, 24 members of TIFY family were identified from the reference genome of *E. ophiuroides*. More specifically, the 24 identified *TIFY* genes were classified into ZML and JAZ subfamilies and were named accordingly. Interestingly, inconsistent with the understanding from most other monocot species, no TIFY subfamily members were detected in *E. ophiuroides*. It should be noted that not all subfamilies of the TIFY family could be found in every plant species, and the TIFY subfamily is also absent not only in the monocots *Brachypodium distachyon* and *Sorghum bicolor* [34,35] but also in the dicot *Camellia sinensis* [33]. A major cause of the TIFY subfamily absence in *E. ophiuroides* might be attributed to the fact that other genes, instead of TIFY subfamily genes, have evolved to function in the roles the TIFY subfamily possessed in other plants [23]. Additionally, another factor worth considering is the possibility of genomic incompleteness. However, in terms of the numbers of TIFY family genes, the *E. ophiuroides* genome encodes an equivalent of or even higher number of *TIFY* genes compared with other diploid grasses such as bamboo (24) [36], *Brachypodium* (21) [34], rice (20) [23], and *sorghum* (19) [37].

An analysis of gene size and structure, conserved domains, and chromosome position is commonly applied to reveal evolutionary clues within a gene family. *TIFY* genes showed considerable variability in gene size and in exon–intron structure in many plant species. In the present study, the member of TIFY family genes in *E. ophiuroides* varies greatly in size and exon number with the shortest gene (*EoJAZ18*) having 204 nucleotides and containing a single exon, while the longest gene (*EoJAZ3*) being 1272-bp long comprising seven exons. More specifically, the exon number of *EoTIFY* genes ranges from 1 to 8, with the fewest number of exons in the subgroup JAZ V and the largest number of exons in the subfamily ZIM/ZML, which is consistent with the findings in many gramineae plants, such as rice [23], *Brachypodium* [34], and maize [24,25], etc. In addition, a similar gene structure and the same distribution of conserved domain patterns were found in the same subfamily, especially the same subgroup, indicating the strong correlation between exon/intron variation and conserved domains and also the significant correlation between the phylogeny and the patterns of exon/intron variation and conserved domains within EoTIFY. According to the extension of exon shuffling theory that the significant correlation between borders of exons and domains creates functional diversity in novel proteins [38], such correlations should play an important role in the evolution of the *EoTIFY* gene family. Moreover, in concert with the findings from many other plants, *EoTIFY* genes were found to be unevenly distributed on six of the nine *E. ophiuroides* chromosomes, which might be related to the expansion of the *EoTIFY* genes driven by gene duplication.

Gene duplication events are the main driving force for the expansion of new gene families and provide an opportunity for novel functions in the evolution of plant genomes [39,40]. Investigation of the occurrence of duplication events contributes to uncover the evolution of genes and species. In the current study, the Ka/Ks ratios (<1) reflected that all of the duplicated gene pairs in the TIFY family of *E. ophiuroides* have suffered an intense purifying selection during the gene evolution. Tandem and segmental duplications are the main pathways of gene duplication [41,42]. In this study, six tandem duplication events and four segmental duplications were detected in the *E. ophiuroides TIFY* gene family. This indicates that both tandem and segmental duplications were predominant duplication events for *E. ophiuroides TIFY* genes, which is in agreement with findings reported in previous research that tandem and segmental gene duplications are the key mechanism for the evolution and expansion of large gene families [42]. 

TIFY proteins have been reported to play important roles in leaf development [3], flower induction [43], and defense responses against abiotic [23,44,45,46] and biotic [12,47,48,49] stresses. In the present study, the expression patterns of *EoTIFY* genes were analyzed in different tissues, and the relatively widespread expression of most of the genes in the leaf and inflorescence was found, reflecting different spatial expression patterns. Similar to the expression of *TIFY* genes in other plant species, the *E. ophiuroides*
*TIFY* genes were highly induced by MeJA. All *E. ophiuroides* TIFY family genes found in this study, except for *EoJAZ2* being downregulated, were upregulated by MeJA at a specific point or during the whole process of MeJA treatment. The stringent consistency of the expression patterns of the three stronger inducible ZML genes, which were downregulated from 1.5 h to 6 h and then tremendously upregulated at 24 h and were grouped to the same subfamily by phylogenetic analysis, may suggest that *EoZML* genes evolved from a common ancestor responsive to the JA signal [50]. Different from the expression patterns of ZML genes, most of JAZ genes were induced and showed a remarkable upregulation in the early stages of MeJA treatment (before 6 h), reflecting the functional conservation of EoJAZ subfamily members associated with JA responses. It is cleared that JA stimulates the degradation of JAZ proteins to derepress transcription factors like MYC2 and MYC3, mediated by the Skp1-Cullin-F-box (SCF) E3 ubiquitin ligase complex SCF^COI1^ [35,51,52]. Therefore, like JAZ genes in many other plant species, the upregulation of these *E. ophiuroides* JAZ subfamily members can be considered to be a fine-tuning method for JA responses [6]. In particular, recent findings reveal that the JA signal can mediate the effect of abiotic stresses and help plants to acclimatize under unfavorable conditions. The JA signal contributes to defend cells from the harmful effects of various environmental stresses [53]. For example, nine *OsTIFY* genes displayed a significant induction in response to JA and all these genes were also induced by one or more abiotic stresses [23]. In addition, in view of the pivotal role of TIFY family genes in response to abiotic stress in many plant species and the strong tolerance of *E. ophiuroides* to Al toxicity, our study mainly focused on the influence of Al stress on the expressions of *EoTIFY* family genes. We noted here that two members of the EoZML subfamily (*EoZML1* and *EoZML2*) and eight EoJAZ genes (*EoJAZ1*, *EoJAZ3*, *EoJAZ5*, *EoJAZ6*, *EoJAZ8*, *EoJAZ9*, *EoJAZ10*, and *EoJAZ21*) were significantly upregulated by Al stress, implying that these ten *EoTIFY* genes may be involved in the regulatory pathway associated with the Al-toxicity response. In this study, ten of the upregulated genes responding to Al stress in *E. ophiuroides* were also induced by exogenous MeJA treatment. This suggests that a JA-dependent signaling pathway might be responsible for regulating the expression of EoTIFY family genes under Al stress.

## 4. Materials and Methods

### 4.1. Identification of TIFY Gene Family Members in the E. ophiuroides Genome

The whole-genome data of the *E. ophiuroides* cultivar (Ganbei) were downloaded from our newly published *E. ophiuroides* genome data [28]. Two basic local alignment search tool (BLAST) methods were used to identify those *TIFY* genes of *E. ophiuroides*. First, 18 AtTIFY and 20 OsTIFY proteins were used as query sequences for the search of *E. ophiuroides* TIFY proteins by BLASTP (E-value < 1 × 10^−5^). The TIFY protein sequences of the model plants *A. thaliana* and *O. sativa* were downloaded from the European Nucleotide Archive (https://plants.ensembl.org/info/data/ftp/index.html, accessed on 15 August 2023). Second, a hidden Markov model (HMM) profile of the conservative functional domain of TIFY (PF06200) was obtained from the Pfam database v35.0 (http://pfam.xfam.org/, accessed on 15 August 2023), and the HMM profile was performed to search for the potential *EoTIFY* gene family members using HMMER package v3.0 software (http://HMMER.org/, accessed on 15 August 2023). Finally, the conserved domain architectures of all candidate gene sequences were further confirmed by a Prosite search (https://prosite.expasy.org/, accessed on 15 August 2023) and the InterPro tool (https://www.ebi.ac.uk/interpro/, accessed on 15 August 2023). Sequences without the typical functional domain of TIFY were excluded from the dataset, and the remaining amino acid sequences containing the conservative TIFY domain were regarded as potential members of the *E. ophiuroides* TIFY family and would be used for subsequent analysis. 

The basic physicochemical properties, mainly including the amino acids number, molecular weight (MW), and isoelectric point (pI), of each *E. ophiuroides* TIFY protein were calculated or predicted using the tool of ProtParam in Expasy (https://web.expasy.org/protparam/, accessed on 15 August 2023). The subcellular localization of barley TIFY proteins was predicted by the web server of the Cell_PLoc 2.0 package (http://www.csbio.sjtu.edu.cn/bioinf/Cell-PLoc-2/, accessed on 15 August 2023).

### 4.2. Sequence Alignment and Phylogenetic Analysis

The multiple sequence alignment (MSA) was performed for TIFY proteins sequences from *E. ophiuroides*, rice, and *Arabidopsis thaliana* using the program of MAFFT v7.487 [54] with auto strategy and default parameter settings. Phylogenetic analysis was carried out by the maximum likelihood algorithm, on the resultant MSAs using IQtree v2.2.0, and the tree topology support was assessed by bootstrap analysis with 1000 replicates. Finally, the phylogenetic tree was annotated and visualized using the online tool of iTOL (https://itol.embl.de/, accessed on 15 August 2023). Furthermore, the sequence logos of the conserved TIFY and Jas functional domains were generated using the web-based application of Web Logo (http://weblogo.threeplusone.com/, accessed on 15 August 2023) [55].

### 4.3. Gene Structure Analysis and Conserved Motif Discovery

The exon/intron organization of *EoTIFY* genes was analyzed using the web server of the Gene Structure Display Server (GSDS) tool [56]. The conserved domain composition of HvTIFY proteins was also analyzed using the Pfam database. In addition, the conserved motif patterns of EoTIFY proteins were identified using the MEME suite v5.5.0 (https://meme-suite.org/meme/tools/meme, accessed on 15 August 2023), with the maximum number of conserved motifs set to 10 [57]. Finally, the gene structure together with the conserved domains and conserved motif of EoTIFYs were examined by TBtools [58].

### 4.4. Chromosomal Locations, Gene Duplication, and Synteny Analysis

The chromosomal position information of *EoTIFY* genes was retrieved from the *E. ophiuroides* genome sequence files and the corresponding gene structure annotation files. *EoTIFY* genes’ localization on the chromosome was visualized by MapChart tools (v2.3.2) [59]. MCScanX analysis was carried out for detecting duplication types, and the collinearity relationship of *EoTIFY* genes was analyzed using the Advanced Circles function in TBtools. The syntenic relationships of the orthologous *TIFY* genes between *E. ophiuroides* and another five species, *Arabidopsis thaliana*, *Brachypodium distachyon*, *Oryza sativa*, *Sorghum biocolor*, and *Setaria italica*, were also analyzed by the MCScanX software (version 1.1.11), and the comparative syntenic map was constructed using the Dual Synteny Plot in TBtools. The genome sequences and general feature format files of the five selected plants were downloaded from the European Nucleotide Archive (https://plants.ensembl.org/info/data/ftp/index.html, accessed on 15 August 2023). The simple Ka/Ks calculator of TBtools was selected to calculate the nonsynonymous (Ka) and synonymous (Ks) substitution values for each duplicated *TIFY* gene pair and the syntenic *TIFY* gene pairs [60].

### 4.5. Plant Materials and Stress Treatments

*E. ophiuroides* seedlings of the strain ‘E039’ were used for qRT-PCR analysis. The seedlings were collected from the Turfgrass Germplasm Resource Nursery at the Institute of Botany, Chinese Academy of Sciences, Jiangsu Province (Nanjing, China). The stolons were taken from the nursery and cut into 4~5 cm pieces each including two nodes and cultivated hydroponically. Based on the *E. ophiuroides* hydroponic culture system, the hydroponic seedlings were pre-cultured for 2~3 weeks to the best growth state, and then used for treatment and harvested. The experiments were conducted in a plant incubator, and the conditions maintained at 30 °C/28 °C (day/night) under a 16 h/8 h (light/dark) photoperiod with 18,000 Lux (~300 µmol·m^−2^·s^−1^). In addition, the callus was induced from the seeds of ‘E039’ as explants, using the modified callus-induction medium (Murashige and Skoog medium, with additive of 2,4-D 2.0 mg/L, 6-BA 0.1 mg/L, proline 0.6 g/L, D-mannitol 20 g/L, sucrose 30 g/L and vegetable gel 3 g/L), and cultured for 30 days in the dark at 25 ± 1 °C.

Before being treated, the root, stem, leaf, inflorescence, stolon node, and callus tissues were harvested (3 biological replicates). The fresh samples were cleaned and frozen in liquid nitrogen immediately. 

As above, the 2~3 weeks pre-cultured *E. ophiuroides* hydroponic seedlings were used for treatment. For the MeJA treatment, the 100 uM MeJA (Solarbio, Beijing, China) solution containing 0.01% Silwet L-77 or the ‘control’ of 0.01% Silwet L-77 aqueous solution was sprayed onto the leaves, respectively (both the MeJA treatment and the ‘control’ were sprayed with twelve biological replicates). Then, the MeJA treated and the control plants were grown in the plant incubator, and the leaves were harvested at 1.5 h, 3 h, 6 h, and 24 h after treatment, respectively (three biological replicates). 

For the Al treatment, the hydroponic seedlings were treated with 1.0 mM Al^3+^ by adding AlCl_3_ into the modified culture solution (1/2 Hoagland culture solution, with the concentration of Pi adjusted to 10 µM, with additive of 1.0 mM CaCl_2_, pH 4.5 ± 1). Then, the Al-treated plants were grown in the plant incubator, and the roots were harvested at 0 h, 6 h, and 24 h after treatment, respectively (three biological replicates).

### 4.6. RNA Isolation and cDNA Synthesis

Total RNA was extracted from various samples with the FlaPure Plant Total RNA Extraction Plus Kit (Genesand Biotech, Beijing, China) with on-column DNase I digestion. The nucleic acid concentration was quantified using an an Epoch spectrophotometer (BioTek, Winooski, VT, USA) at 260 nm; meanwhile, the 260/280 nm ratio within the range of 1.80~2.20 and a 260/230 nm ratio of approximately 2.00 were retained. The cDNA was synthesized from 1 µg of total RNA using random hexamers with the HiScript III 1st Strand cDNA Synthesis Kit (Vazyme, Nanjing, China).

### 4.7. Primer Design and qRT-PCR Analysis

According to the *TIFY* genes identified from the genome data of *E. ophiuroides* [28], specific primers for the genes were designed using the software PRIMER PREMIER 5.0 with the following parameters: primer length of 19~25 bp, GC content 40~60%, melting temperature 55~65 °C, and amplicon length 120~350 bp (Appendix A). In addition, we used the ‘*CACS*’ gene as the reference gene for treated root samples and the ‘*PP2A*’ gene as the reference gene for the treated leaf samples, respectively, which was previously identified and selected in *E. ophiuroides* [61]. qRT-PCR was performed on a Jena qTower3 platform (Analytik Jena AG, Jena, Germany) and carried out in a 20 µL reaction mixture using the ChamQ Universal SYBR qPCR Master Mix (Vazyme, China). The thermal cycling program was: 95 °C for 30 s, and 40 cycles of 95 °C for 10 s and 60 °C for 30 s, followed by a melt curve step. All reactions were performed in three replicates. The relative quantitative method [62] was used for the Ct data analysis. The relative quantitative 2^−ΔΔCt^ data were analyzed by IBM Statistical Package of Social Sciences (SPSS) version 19.0 (SPSS, Inc., Chicago, IL, USA).

## 5. Conclusions

In this study, the identification of and characterization of the *TIFY* genes in *Eremochloa ophiuroides* were performed at the genome-wide level. In total, 24 *EoTIFY* genes were detected and could be classified into the ZIM/ZML and JAZ subfamilies, with the JAZ subfamily being further divided into five subgroups (JAZ I to JAZ V). EoTIFY family genes showed great variation in the number of exons and the length of introns among different subfamily and subgroup members. Six tandem and four segmental duplication events were detected in the EoTIFY family. The constructed collinear maps between *E. ophiuroides* and five representative species including four monocot and one dicot plants provided valuable clues for understanding the potential evolution of the EoTIFY family. Importantly, the *TIFY* genes of *E. ophiuroides* were strongly induced by MeJA. Ten *EoTIFY* genes (*EoZML1*, *EoZML1*, *EoJAZ1*, *EoJAZ3*, *EoJAZ5*, *EoJAZ6*, *EoJAZ8*, *EoJAZ9*, *EoJAZ10*, and *EoJAZ21*) showed high expression levels under both exogenous MeJA treatment and aluminum stress, indicating that the JA-dependent signaling pathway might be associated with the response of *EoTIFY* genes to aluminum toxicity. This study provides a rational basis for further understanding the function of *TIFY* genes in *E. ophiuroides*, as well as useful information and reference points for mining genes related to aluminum tolerance in warm-season grass plants.

## Figures and Tables

**Figure 1 plants-13-00462-f001:**
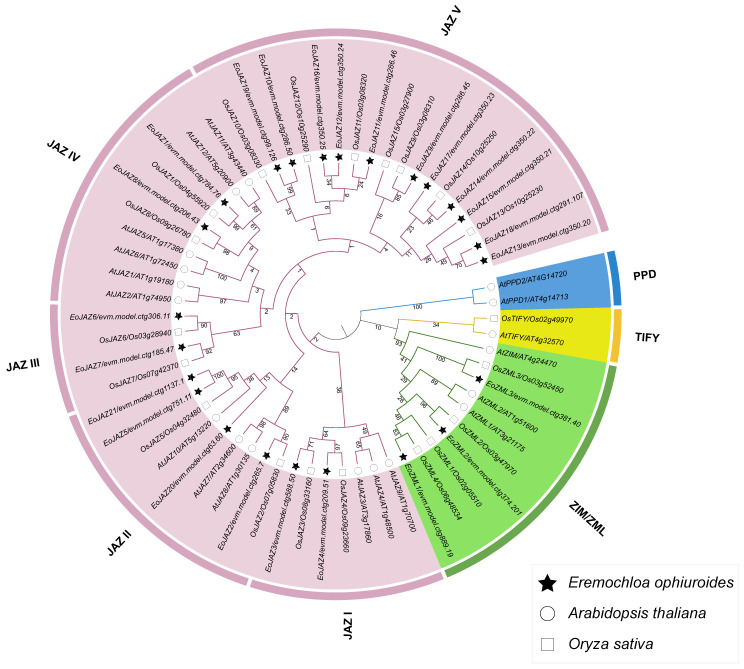
Phylogenetic relationships of the TIFY proteins from *E. ophiuroides*, *Arabidopsis*, and *O. sativa*.

**Figure 2 plants-13-00462-f002:**
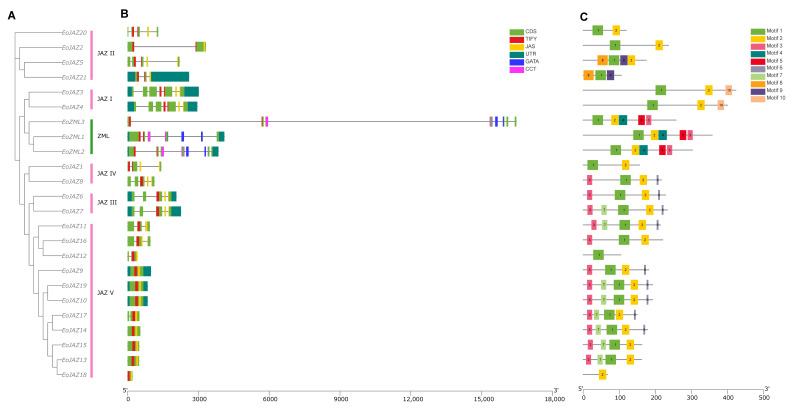
Phylogenetic relationships, gene structure, and conserved motif of the *E. ophiuroides TIFY* gene family. (**A**) The phylogenetic relationships of the *E. ophiuroides* TIFY family. (**B**) Exon–intron structure and the characteristic domain distribution of the *E. ophiuroides TIFY* genes. (**C**) Distribution of the conserved motifs in *E. ophiuroides* TIFY proteins.

**Figure 3 plants-13-00462-f003:**
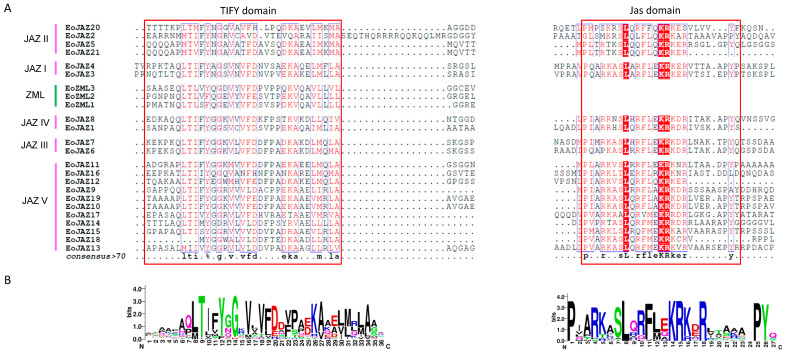
Alignments of TIFY and Jas domain sequences and the sequence logos of the TIFY family members in *E. ophiuroides*. (**A**) Multiple sequence alignments of *E. ophiuroides* TIFY family members. The conserved TIFY or Jas domain is boxed in red, and the red background colors in the boxes indicate the sequence consistency of 100%, while the blue squares indicate the sequence identity above 75% with the conserved sequence symbols displayed in red font. (**B**) Sequence logos of TIFY and Jas domains.

**Figure 4 plants-13-00462-f004:**
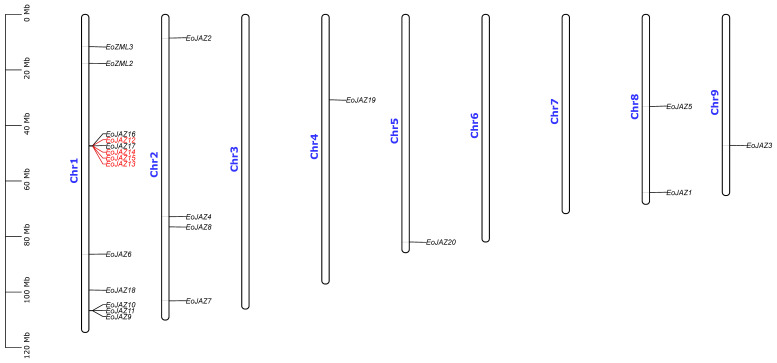
Distribution of the *EoTIFY* genes on *E. ophiuroides* chromosomes. The red font represents the tandem gene duplication. The chromosome numbers are indicated at the left side of each chromosome image. The scale on the left is in million bases (Mb).

**Figure 5 plants-13-00462-f005:**
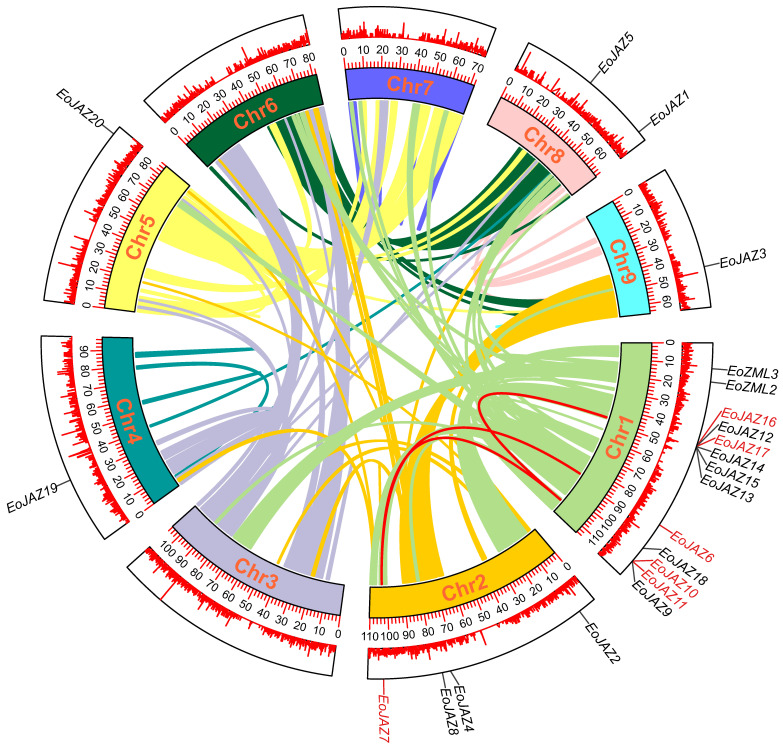
Colinear region of the *E. ophiuroides TIFY* genes. The colored lines represent all the colinear blocks in the *E. ophiuroides* genome, and the red lines represent *TIFY* gene pairs subjected to segmental duplication. Chromosome numbers are shown at the center of each chromosome.

**Figure 6 plants-13-00462-f006:**
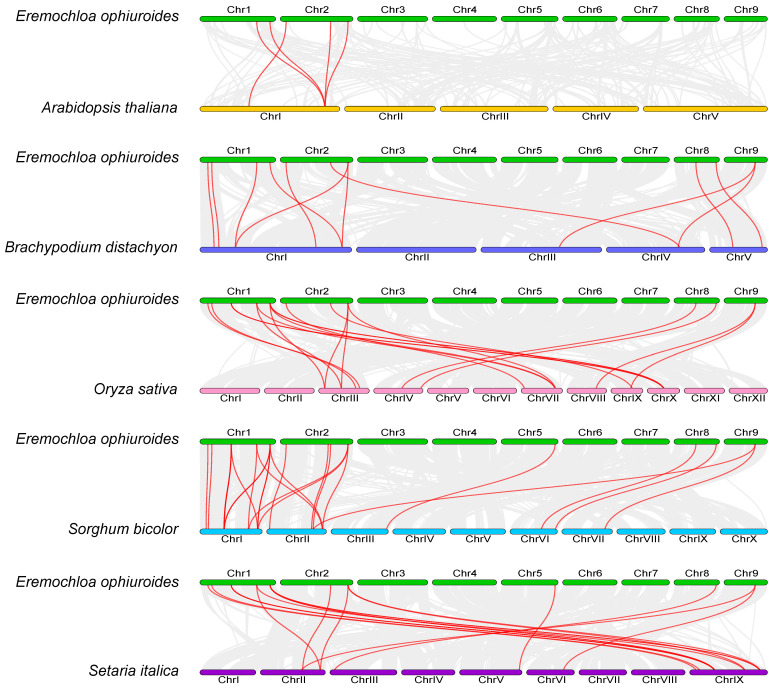
Synteny analysis of *TIFY* genes between *E. ophiuroides* and *Arabidopsis thaliana*, *Brachypodium distachyon*, *Oryza sativa*, *Sorghum bicolor*, and *Setaria italica*. Gray lines in the background indicate the collinear blocks within *E. ophiuroides* and other plant genomes, while the red lines highlight the syntenic *TIFY* gene pairs.

**Figure 7 plants-13-00462-f007:**
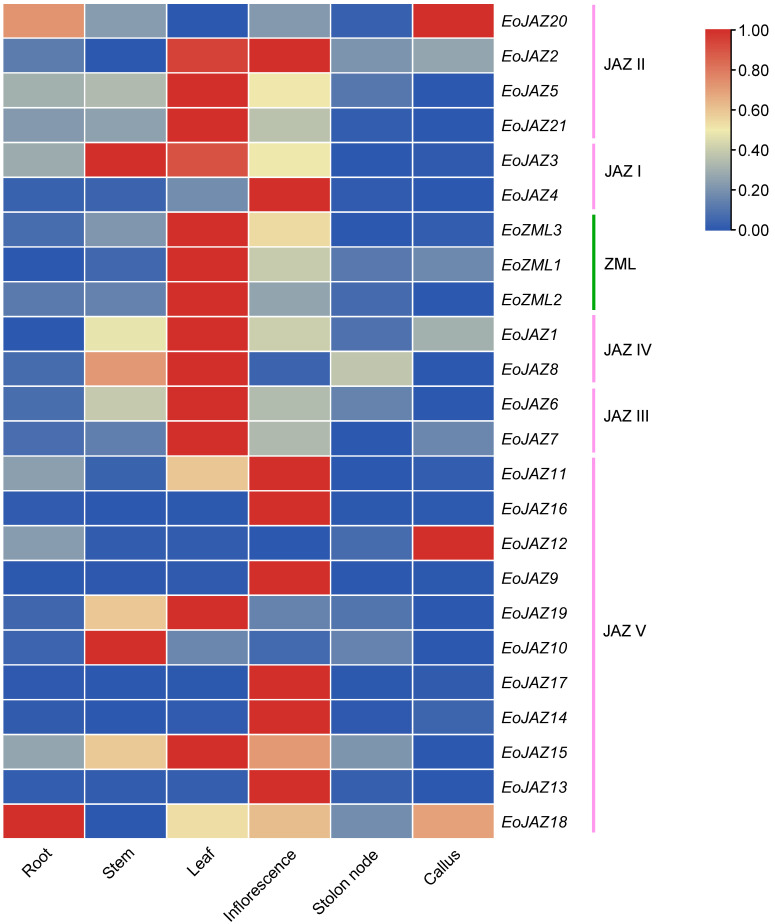
Expression patterns of the *EoTIFY* family genes in various tissues of *E. ophiuroides*.

**Figure 8 plants-13-00462-f008:**
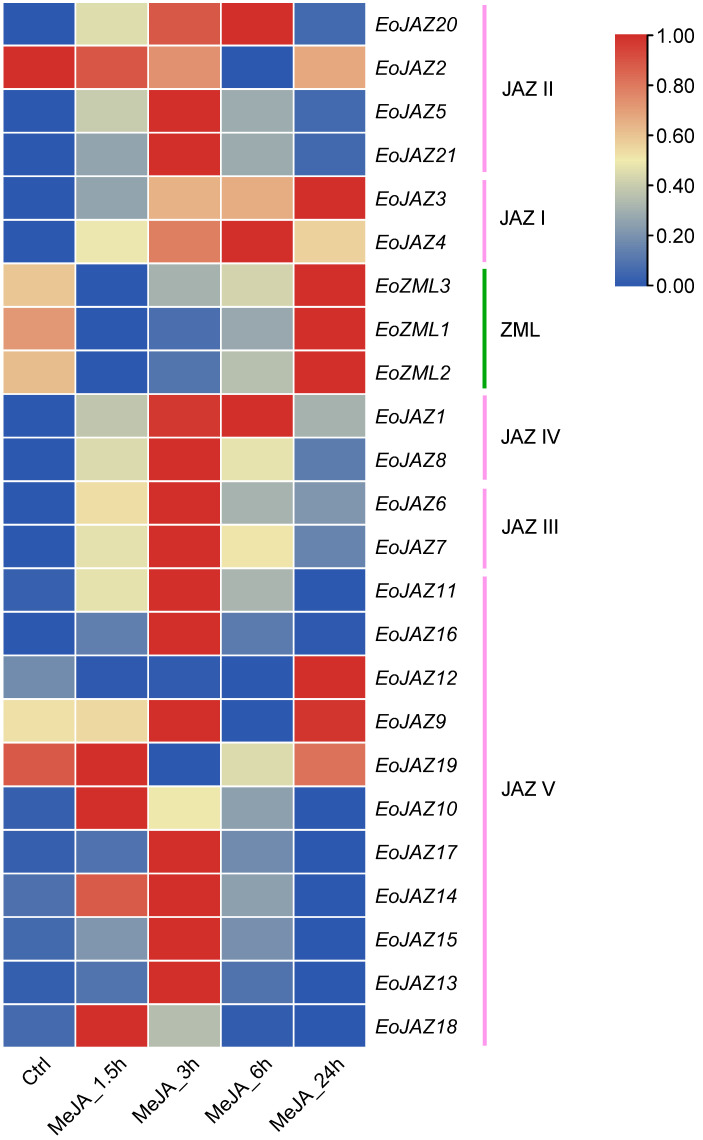
Expression patterns of the *EoTIFY* family genes under MeJA treatment.

**Figure 9 plants-13-00462-f009:**
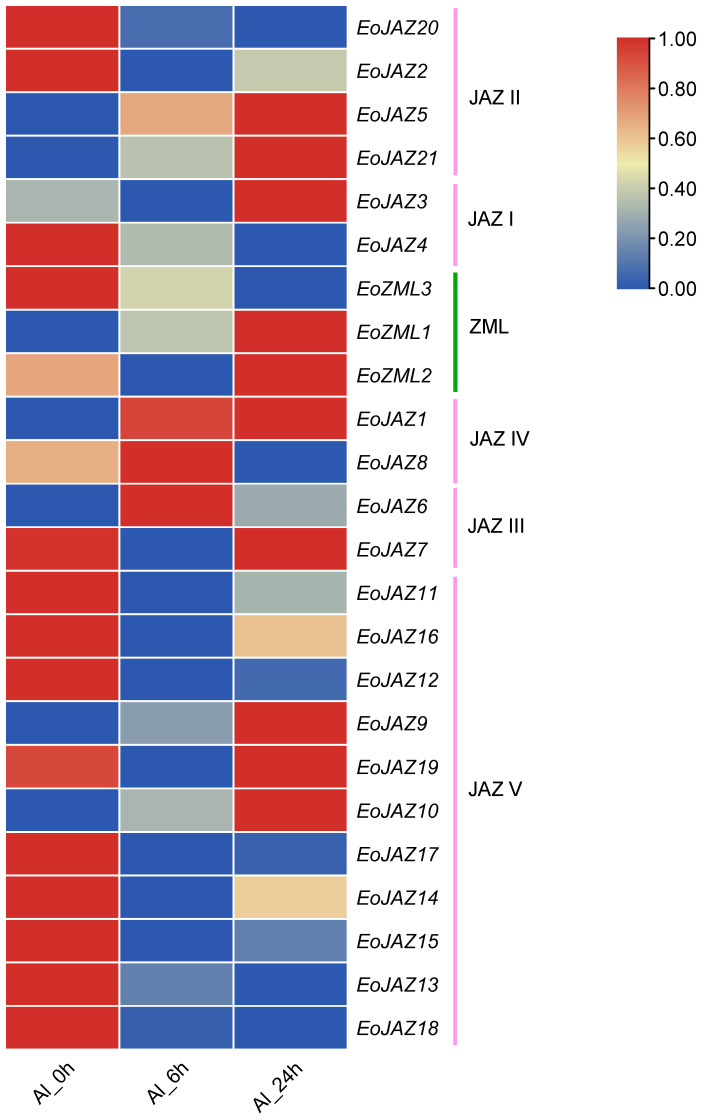
Expression patterns of the *EoTIFY* family genes in response to aluminum toxicity.

## Data Availability

The original contributions presented in the study are included in the article/Appendix A, further inquiries can be directed to the corresponding author.

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
