# Peer review of "Genome-Wide Identification and Characterization of the *TIFY* Gene Family and Their Expression Patterns in Response to MeJA and Aluminum Stress in Centipedegrass (*Eremochloa ophiuroides*)"

_plants, 2024, doi:10.3390/plants13030462_

Round 1
Reviewer 1 Report
Comments and Suggestions for Authors
The present research has studied the genome wide identification of the TIFY genes on Eremochloa ophiuroides in response to Aluminium stress.
The present document is well written porvides both in the introduction and the discussion a really good litterature review, the abstract is pertinent and adecuate to the research results.
The introduction present the research question justified and clealy.
The results are well described and the figures associated are clear and well presented.
The discussion is complete and well references.
Globally the material and method is complete few data have to be add to be perfect. See attached document for the details;
The conclusion is coherent to the results

Reviewer 2 Report
Comments and Suggestions for Authors
In the current study authors indenitified and chaarcterized 24 TIFY genes in in warm-season grass plants Eremochloa ophiuroides.
There are some points need to be corrected.
Line 17: „factors involved in regulating growth, development, signal transduction and stress responses in plants.“ – grammarly.
Line 22: „infertile soils“ ?.
Lines 32-33 : do you mean treatment with exogenous? Have you tried other hormones ?
Ine 35: „were observed to exhibit“ ??? “may contribute to a role“ ¿?
Line 52: acid soil and aluminium have a direct link. Authors should describe this in details. Line 72: „regulators regulating grain size“??
Line 258: „significantly tissue specific expression”?
Line 270 -272: please, provide details of callus types, what was hormone ttreatments.
Lines 274 – 319: you mention that TIFY have a different expression level dependent form cell type. But in this part you describe only leaves. So, this data need to be taking into account with precaution.
Please, describe subcellular localization in different cell type. Tis is the most importnat for gene functions.
Line 278: „Thereupon“ ??
Line 482: plesae, provide light in µmol.
Lines 485- 486: please re-write.
Comments on the Quality of English LanguageCheck carefully all text and made corrections.
Reviewer 3 Report
Comments and Suggestions for Authors
Manuscript "Genome-wide identification and characterization of the TIFY gene family and their expression patterns in response to MeJA and aluminum stress in centipedegrass (Eremochloa ophiuroides)" is very interesting.
General comments:
Authors investigated the potential roles of E. ophiuroide TIFY proteins in response to phytohormone and Al stress.
Authors analysed gene structures, alignment of conserved domains, phylogenetic analysis, investigations of chromosomal location and syntenic relationships for all of the identified E. ophiuroides TIFY genes.
Authors examined expression patterns of the E. ophiuroides TIFY genes in different tissues, response to phytohormone MeJA treatment and Al toxicity. These results provide a foundation for further functional research of TIFY family members in E. ophiuroides, thus facilitating the cultivation of Al-resistant cultivars in grass plants.
Detailed comments:
The introduction is written very well.
The results and M&Ms are very extensively described. Unfortunately, a very important aspect of all scientific articles, namely statistical analysis, was missing.
Line 515: The authors wrote that all reactions were carried out in triplicate. Unfortunately, no statistical analysis was performed. In the M&M and in the captions to Figures 1 and 2, it is not stated what measure of similarity calculation was used.
The quality of Figure 2 is very poor. It should be improved.
The authors report the expression patterns of EoTIFY family genes in different tissues of E. ophiuroides, but unfortunately do not compare them among themselves. Appropriate statistical methods should be used, because at the moment it is only a data submission without drawing appropriate conclusions.
The same is true for 2^-ΔΔCt values.
Paper needs major revision.
Round 2
Reviewer 2 Report
Comments and Suggestions for Authors
The authors made a great work, Some minor editing still reguire for the publications.
Comments on the Quality of English Language
Minor grammar/
Author Response
We have revised the introduction, results and discussion sections in the revised manuscript. Meanwhile, we upload high-quality figures as zip. Of course, the text has also been polished in English writing.
Reviewer 3 Report
Comments and Suggestions for Authors
To my point about the lack of indication of the similarity measure used in the manuscript, the authors indicate "maximum likelihood." This response demonstrates a lack of knowledge of the analyses used in the manuscript. "Maximum likelihood" is an algorithm for constructing dendrograms, not a measure of similarity. The authors unknowingly use packages without knowing the basics or someone else performed these analyses.
In such a situation, I do not see how this manuscript can be accepted for publication.
Author Response
To reviewer # 3:
Comment 1: “To my point about the lack of indication of the similarity measure used in the manuscript, the authors indicate "maximum likelihood." This response demonstrates a lack of knowledge of the analyses used in the manuscript. "Maximum likelihood" is an algorithm for constructing dendrograms, not a measure of similarity. The authors unknowingly use packages without knowing the basics or someone else performed these analyses. In such a situation, I do not see how this manuscript can be accepted for publication.”
Answer: We’re sorry for our previous misunderstanding of the reviewer’s comments. As for reviewer’s comment about the measure of similarity calculation, our explanation is as follows: As we described in M&M, we performed multiple sequence alignment (MSA) using MAFFT software with auto strategy and default parameter settings, and the specific operating command under Linux was “ ~ ../../mafft -- auto At_Os_Eo_MSA_analysis.fa > ../MSA_ At_Os_Eo_mafft.output”. For MAFFT, when the participating alignment sequence is less than 200 sequences, it defaults to selecting the L-INS-i algorithm, which is an iterative refinement method using the weighted sum-of-pairs (WSP) and consistency scores. Specifically, L-INS-i has advantage for aligning a set of sequences containing sequences flanking around one alignable domain. Hence, in our case of MSA analysis of 62 sequences with a conserved TIFY domain, it was a good choice to select MAFFT strategy. Finally, we sincerely hope that the reviewers can have friendly exchanges with us regarding some academic issues, and we are very willing to accept guidance from peer experts.

Round 3
Reviewer 3 Report
Comments and Suggestions for Authors
The answer given by the Authors once again shows a lack of knowledge in writing scientific articles. Once again, I repeat that the description of the methods used must be presented in such a way that anyone can repeat it. Providing the software used is not a methodology. A scientist who would like to carry out an analogy using other software will not learn from a peer-reviewed manuscript what methods were used.
I encourage the authors to read the software manual and list in the description the methods that are used during each action item.
As it stands, the manuscript is not suitable for publication.